# RETRACTED: Magnetic Ionic Liquid Nanocatalyst to Improve Mechanical and Thermal Properties of Epoxy Nanocomposites

**DOI:** 10.3390/nano10122325

**Published:** 2020-11-24

**Authors:** Ayman M. Atta, Hamad A. Al-Lohedan, Ahmed M. Tawfeek, Nourah I. Sabeela

**Affiliations:** Department of Chemistry, College of Science, King Saud University, Riyadh 11451, Saudi Arabia; hlohedan@ksu.edu.sa (H.A.A.-L.); a.tawfik@ksu.edu.sa (A.M.T.); noonah-37@hotmail.com (N.I.S.)

**Keywords:** magnetite, imidazolium ionic liquid, epoxy, curing, dynamic mechanical, thermal

## Abstract

New magnetic imidazolium ionic liquid (IIL) was synthesized to improve the curing, mechanical, and thermal characteristics of the epoxy/polyamine system. In this respect, 2-(4-minophenyl)-1.3-bis(triethoxysilyl)-1H-imidazol-3-ium acetate as IIL was synthesized and characterized by different spectroscopy tools. The IIL was used as capping to prepare Fe_3_O_4_ nanoparticles (NPs) as new Fe_3_O_4_-IIL NPs. The thermal stability, morphology, crystal lattice structures, and magnetic properties were evaluated to confirm the formation of uniform, thermal, stable, and superparamagnetic Fe_3_O_4_-IIL NPs. The prepared Fe_3_O_4_-IIL NPs were mixed with an epoxy/polyamine system to improve the curing, thermal, and mechanical properties of epoxy through chemical reactions. The dynamic mechanical analyzer and differential scanning calorimeter were used to investigate the flexibility and storage modulus of the cured epoxy/polyamine system in the absence and presence of Fe_3_O_4_-IIL NPs. The atomic force microscope and scanning electron microscope were used to evaluate the dispersion and embedding of Fe_3_O_4_-IIL NPs into epoxy matrix. The thermal, mechanical, and surface morphologies data confirmed that the incorporation of Fe_3_O_4_-IIL NPs using 3 wt. % during the curing of an epoxy/polyamine system produces superior epoxy films without cracks, holes, and NPs agglomeration.

## 1. Introduction

Epoxy thermoset polymers have great industrial applications as structural composites, adhesive, and organic coatings [1]. The mechanical properties of the cured epoxy resins were affected by the curing rate, degree of crosslinking, hardener chemical structures, fillers types, and contents [2]. The modification epoxy chemical structures and degree of crosslinking were investigated to obtain useful mechanical properties such as high modulus, strength, and glass transition temperature, although they produced tough, brittle, and cracked surfaces due to fast curing [1]. Moreover, the higher rigidity and brittleness degree of the cured epoxy films decrease their resistivity for cracking propagation to influence the mechanical, barrier, anti-corrosive, and coatings performances of the cured epoxy films as external coatings [2]. Usually, there are two techniques based on improvement: the epoxy chemical curing and physical blending of fillers with epoxy networks were used to improve their anticorrosion, mechanical, adhesion, and thermal properties [3,4,5]. In this respect, wide verities of chemicals were used to tailor and control the chemical crosslinking of epoxy via forming chemical bonds [4,5,6]. The modification of fillers surfaces with either epoxy or amines functional groups was better to embed the fillers with the epoxy networks via chemical linking [4,5,6]. This technique was used to produce multifunctional epoxy organic coatings with superior coating performances [3,4,5,6]. However, the nanomaterials attracted great attention to overcome the limitation of polymeric and inorganic filler materials via increasing the epoxy thermal and mechanical properties using lower concentrations of multifunctional nanofillers [7,8,9,10,11,12]. The nanomaterials were used to improve the fracture toughness of epoxy resins [13,14,15]. There are two main challenges influencing the formulation and applications of nanomaterials with polymers to produce polymer nanocomposites with superior properties. One is based on the dispersion of nanomaterials in the polymer nanocomposites, and the other is the nature of the interactions between polymers chains or networks with nanomaterials. The aggregation of nanomaterials decreases their dispersion in the epoxy or hardener resins during the curing and increases the epoxy processing viscosity to limit their applications as nanofillers [13,14,15].

Recently, ionic liquids (ILs), as organic salts at room temperature, have been used to improve either the curing or processing of epoxy resins with superior properties and to prolong the pot life of epoxy compositions [16,17,18,19,20]. Moreover, the ILs were used to prepare nanomaterials and control their sizes, shapes, and dispersion [21,22]. The ILs nanocomposites are proposed to apply as superior nanofillers to improve the curing of epoxy resins having excellent mechanical, thermal, and anti-corrosion properties [23,24]. In this respect, the objective of this work is to modify the nanomaterials surfaces with new ILs that contain an amine functional group to chemically bond with epoxy networks during the curing process to improve its curing, epoxy network crosslinking densities, and thermomechanical performances. The chemical structures of imidazolium ILs anions (organic cations and ions) were varied to investigate their reactivity as both capping for nanomaterials and a curing agent with the epoxy resins [25,26]. Imidazolium ILs (IILs) can attack or activate the nucleophilic reaction on the oxirane ring of epoxy resins to control the crosslinking densities of the polymer networks. In this respect, imidazolium ILs were used to improve the mechanical properties and to decrease of the onset curing temperature of epoxy. ILs were used as a curing agent to offer both good stability of the ring in oxidative and low viscosity formulations [25,26]. The magnetite nanoparticles (NPs) have attracted great considerable research interest due to their unique catalytic, barrier properties as anticorrosive and toughening efficiency improver epoxy films [7,27]. IILs were previously used to prepare highly stable magnetic IILs nanodispersions using a silanization process to form surface-modified magnetic IILs [28]. IILs provide a flexible liquid platform for catalysis by transition metal NPs (MNPs) as a stabilizer, ligand capping, and support for MNPs [29]. In our previous work [30,31], silylated IILs were prepared and used to produce mesoporous silica/magnetite nanoparticles as effective adsorbents. In this respect, silylated IIL can be designed as curing and dispersing agents for magnetite NPS to improve thermal and mechanical properties of epoxy. Herein, the present work aims to design IILs combined with siloxane and inorganic magnetic nanoparticles to improve the dispersion, magnetic, and oxidative stability of magnetite NPs to act as nanofiller. Moreover, the amine functional group on the IIL surface was used to promote the curing of epoxy and chemical link both IIL and magnetite NPs with the epoxy matrix. In this respect, 3-aminopropyltriethoxysilane (APTS) was condensed with 4-aminobenzaldehyde (ABA) in the presence of glyoxal to produce IIL. The siloxyimidazolium IL was hydrolyzed with tetraethoxysilane (TES) and iron cations (Fe^3+^ and Fe^2+^) in the basic medium to produce magnetite IIL. It is expected that the incorporation of magnetite IIL during the curing of the commercial epoxy resin with polyamine (PA) hardener will alter the densely crosslinked part of the epoxy networks and their curing mechanism due to the presence of imidazolium, amine, and hydroxide functional groups on magnetite NPs. For these reasons, the curing of epoxy resin with PA in the presence and absence of magnetite IIL was investigated by differential scanning calorimeter and dynamic mechanical analyses. The improvement of molecular crosslinking of the epoxy networks with magnetite IIL is another goal to enhance the adhesion, mechanical, and thermal properties of epoxy resins.

## 2. Experimental

### 2.1. Materials

All chemicals consumed in this work were supplied from Sigma-Aldrich Chemicals Co. and used without further purification due to their high analytical grade. Glyoxal (GA), 4-aminobenzaldehyde (ABA), and acetic acid were used to produce imidazolium IL (IIL). Anhydrous FeCl_3_ and KI and ammonia solution (25 wt. %) were used for magnetite IIL synthesis (Fe_3_O_4_-IIL). Tetraethoxysilane (TES) and 3-aminopropyltriethoxysilane (APTS) were used as siloxane precursor sources. Deionized DIW (DIW) with 0.1 MΩ cm resistivity was used for preparing IIL. Epikote epoxy resin 828 (based on bisphenol A diglycidyl ether; DGEB) with epoxy equivalent weight 190–200 g/eq (Hexion, Olana, Italy) and its long-chain length aliphatic polyamines Epikure™ 270 hardener (PA) were used to prepare epoxy network after mixing with weight ratio 4:1 (wt. %).

### 2.2. Preparation of Magnetic IIL

#### 2.2.1. Preparation of IIL

APTS (0.5 mol; 11.1 g) dissolved into acetic acid aqueous solution (50 mL; 50/50 vol. %) was added to the mixture of GA (0.025 mol; 1.45 g), ABA (0.025 mol; 3.025 g), and acetic acid aqueous solution (50 mL; 50/50 vol. %) at temperature −4 °C. The reaction mixture was stirred (900 rpm) for 30 min until clear solution was obtained. The reaction temperature was increased up to 70 °C (heating rate 2 °C per 5 min) and kept for 5 h, after which it was cooled for purification. The reaction mixture was washed several times (five times) with diethyl ether to remove the unreacted materials and to obtain a colorless organic phase. The unreacted GA, acetic acid, and DIW were removed by heating under reduced pressure (35 °C at 40 psi) for 24 h using rotary evaporator to obtain colorless liquid of IIL with 95% yield percentage. The nitrogen content of the purified IIL was determined using (Kjeldahl method; using a Tecator Digestion System (Hilleroed, Denmark) and a Tecator distillation unit, Kjeltec 1003), and found to be 8.245 ± 0.07 wt. %, which agrees with the theoretical value (8.25 wt. %).

#### 2.2.2. Preparation of Fe_3_O_4_-IIL

Anhydrous FeCl_3_ solution (4 g; dissolved in 30 mL DIW) was added to KI solution (1.32 g, dissolved in 5 mL DIW) and stirred using a mechanical stirrer (400 rpm) at room temperature under N_2_ atmosphere for 1 h. The iron cations filtrate was obtained after removal of the solids precipitate (iodine byproduct) by filtration as described in the previous work [30]. IIL (1 g) and TES (1 g) were dispersed into DIW/ethanol mixture (100 mL; 50/50 vol. %) using an ultrasonic processor (TEC-40 model, Roop-Telsonic Ultrasonics Ltd., Mumbai, India; power density, 750 watts; frequency, 20 kHz) for 5 min. The IIL and TMS solution was added to the iron cations solution at the same time with ammonia solution (10 mL; 25 wt. %) under vigorous mechanical stirring (1200 rpm). The reaction mixture was heated and stirred at 45 °C for 36 h. The magnetic IIL was collected from the reaction suspension by an external magnet and dispersed into HCl solution (1 L; 4 M) for 1 h to remove the uncapped magnetite. The Fe_3_O_4_-IIL was separated from the suspension by an external magnet (made from the neodymium magnet; magnetic attraction force, 845.8 N; magnetic flux density, 534 mT). The precipitate was washed several times (5 times) with ethanol. The yield percentage of reaction was 90 wt. %.

### 2.3. Characterization of Magnetite and IIL

The chemical structures of IIL and Fe_3_O_4_-IIL were investigated by using Fourier transform infrared analysis (Nicolet Magna 750 FTIR spectrometer using KBr, Newport, NJ, USA). The hydrogen and carbon nuclear magnetic resonance (^1^HNMR and ^13^CNMR; 400 MHz Bruker Avance DRX-400 spectrometer; Toronto, ON, Canada) was carried out using deuterated dimethyl sulfoxide (DMSO) as organic solvent and tetramethylsilane as internal solvent. The particle size, polydispersity index (PDI), and zeta potential of aqueous dispersion of Fe_3_O_4_-IIL in the presence of 0.001 M KCl at 25 °C were measured using dynamic light scattering (DLS; Malvern Instrument Ltd., London, UK). The DLS calibration was carried out for standardized aqueous solutions having particle sizes from 1 to 100 nm and zeta potential ranged from −40 to 50 mV. The crystalline lattice structure of Fe_3_O_4_-IIL was determined using X-ray powder diffraction (X’Pert, Philips, Amsterdam, The Netherlands, using CuKa radiation of wavelength λ = 1.5406 Å with 40 kV voltage and 35 mA intensity). XRD patterns of Fe_3_O_4_-IIL were recorded in the range of 15° ≤ 2*θ* ≤ 80° at the scan speed of 0.01° s^−1^. The magnetic characteristics of Fe_3_O_4_-IIL were measured using a vibrating sample magnetometer (VSM; USALDJ9600-1; LDJ Electronics, Troy, MI, USA) operating at a vibrational frequency of 75 Hz. A magnetic hysteresis loop of Fe_3_O_4_-IILwas recorded at room temperature under an applied magnetic field that ranged from −20,000 to +20,000 Oe to determine its saturation magnetization (Ms). The thermal stability of IIL, Fe_3_O_4_-IIL, and cured DGEB/PA films was evaluated using thermogravimetric and differential thermogravimetric analysis (TGA-DTG; NETZSCH STA 449 C instrument, New Castle, DE, USA) under an N_2_ atmosphere with a heating rate of 10 °C min^−1^ and flow rate of 60 mL min^−1^. Weights of samples used in this experiment were between 10 and 20 mg. The morphologies of Fe_3_O_4_-IIL were evaluated by transmission electron microscopy (TEM JEOL JEM-2100 F) at an acceleration voltage of 200 kV JEOL, Tokyo, Japan) and scanning electron microscopy (JEOL JXA-840A) at 10 kV. The Fe_3_O_4_-IIL dispersion in ethanol was sprayed over the 3 mm TEM carbon-coated copper grid and dried overnight at room temperature (25 ± 2°C). The repeatability precision for all IIL and Fe_3_O_4_-IIL samples analyses was successively analyzed 6 times. Three different sample weights of certified reference material, analyzed in three replicates, were used for trueness determination. The range of linearity was evaluated by checking the linear regression coefficient (*R*^2^) of the calibration curve. The linearity of the calibration curve was considered acceptable when *R*^2^ > 0.995.

### 2.4. Curing of DGEB/PA in the Presence of Fe_3_O_4_-IIL

The thermal characteristics of IIL and Fe_3_O_4_-IIL were determined by using differential scanning calorimetry (DSC; Q10 DSC calorimeter from TA Instrument). All samples were dried in a vacuum oven for 2 h at 45 °C to remove any contaminated air humidity. A DSC device equipped with manual cooling unit and calibrated with zing was used to determine the phase transitions of IIL and Fe_3_O_4_-IIL (10–20 mg) after they were sealed in an aluminum pan. The samples were subsequently heated to 80 °C after they cooled to −120 °C under a heating rate of 5 °C min^−^^1^ under a N_2_ flow. The glass transition temperature (T_g_) and the heat evolved during the curing exothermic reaction DGEB/PA (ΔH) in the presence and absence of Fe_3_O_4_-IIL were measured by a DSc device. In this respect, different amounts of Fe_3_O_4_-IIL (1–6 wt. % related to the total weight of DGEB and PA resins) were dispersed with PA using a mechanical stirrer at a speed of 1000 rpm and ultra-sonicating with alternative 30 s cycles using ultrasonic waves of 20 kHz (Bandel in Co., 150 W). The PA suspensions were manually mixed with the recommended DGEB weight percentages (PA: DGEB; 1:4 wt. %) into a Pyrex beaker put in an ice bath. The sample of DGEB/PA (5–7 mg) in the absence or presence of Fe_3_O_4_-IIL was sealed in hermetic aluminum pans and an identical empty reference pan was used to analyze their curing characteristics by DSC analyzer. The curing exothermic was evaluated by using the non-isothermal DSC measurements. Then, the sample pan was put in the DSC cell, which was pre-cooled to −50 °C. The DSC cell was subsequently heated at a constant rate 5 °C min^−1^ from −30 to 300 °C under N_2_ atmosphere. The ΔH was evaluated by measuring the integrating the exothermic peak. Three measurements were carried out for each sample to determine the averages of the measurements.

Dynamic mechanical properties of the cured DGEB/PA in the presence of Fe_3_O_4_-IIL were evaluated using a dynamic mechanical analyzer (DMA; Q200, TA Instruments) in double cantilever mode. The DGEB/PA was cured and casted into Teflon molds and hardened at 150 °C/2 h as rectangle-shaped samples having the dimensions of the 20.0 × 10.0 × 5.0 mm^3^. The samples were cooled and heated from 0 to 300 °C at a rate of 3 °C min^−1^ under N_2_ atmosphere with a frequency of 1 Hz and amplitude of 40 μm. The T_g_ and tan δ values were determined using DMA with a dual cantilever at a heating rate of 2 °C min^−1^ from 30 to 220 °C at a frequency of 1 Hz.

The ultra-thin sectioning (50–70 nm) of the cured DGEB/PA in the presence of Fe_3_O_4_-IIL films was performed by ultra-microtomy at low temperature. The morphology of the fractured surfaces of the samples was investigated using scanning electron microscopy (JEOL JXA-840A). Atomic force microscope (AFM; Agilent 5500 with multipurpose closed loop scanner) is used to investigate the surface roughness and topologies of the cured DGEB/PA using different weight ratios of Fe_3_O_4_-IIL. Imaging the DGEB/PA was performed in the tapping mode with silicon cantilevers with a nominal spring constant of 48 N/m and a resonance frequency of around 300 kHz. The scanning rates and resolution were 12 Hz and 512 pixels per line, respectively.

## 3. Results and Discussion

The IIL can be easily prepared by the condensation of the primary amine with aldehyde under acidic condition [30]. In this respect, ABA was selected as the aromatic aldehyde to condense with APTS as the primary amine in the presence of GA under acidic condition to prepare 2-(4-minophenyl)-1.3-bis(triethoxysilyl)-1H-imidazol-3-ium acetate as IIL, as illustrated in the experimental section and Figure 1. In the previous works [30], p-hydroxybenzaldehyde was used as the aromatic aldehyde that affected the solubility of the produced IIL. In this work, the amine group of ABA facilitates its condensation with APTS and the solubility of the produced IIL (Figure 1) due to the formation of amine salt in the presence of acetic acid. The formation of amine salt of ABA (CH_3_COOH.NH_2_–Ph) at temperature (−4 °C) prevents the condensation of the amine group with GA and facilitates the condensation of its aldehyde group with APTS. The produced IIL (Figure 1) was easily mixed with ethanol and water as well as toluene and xylene. The IIL was added to iron cations, prepared from the reaction of FeCl_3_ with KI after the removal of I_2_, under the basic condition to hydrolyze the triethoxysilyl groups of IIL with TES. The siloxane as well as formation of iron oxide NPs can be formed according to the sol–gel technique and co-precipitation method [31]. The proposed chemical structures of IIL and Fe_3_O_4_-IIL (Figure 1) without the formation of other iron oxides as well as the thermal stability, crystalline lattice structure, surface morphologies, and magnetic properties will be confirmed in the forthcoming section.

### 3.1. Characterization of the Prepared IIL and Fe_3_O_4_-IIL

The purity and chemical structure of the prepared IIL were identified from its ^1^HNMR and ^13^CNMR spectrum represented in Figure 1a,b, respectively. The all ^1^HNMR peaks of IIL were marked and correlated to their corresponding protons (Figure 1a). The appearance of deshielded imidazolium and methyl protons appeared at 7.7 and 1.6 ppm, respectively, confirms the formation of imidazolium cation and an acetate ion. The appearance of the broad peak at 5.5 ppm of amino group elucidates that the amine group of ABA was not contributed in the condensation with GA due to it weaker basicity than APTS. The appearance of peaks at 0.6 and 3.53 ppm correlated to methyl and methylene protons, respectively, indicates that ethoxy groups of APTS were not hydrolyzed under acidic condition. The ^13^CNMR spectrum of IIL (Figure 1b) elucidate the formation of imidazolium cation and acetate anion with the appearance of peaks at 144 and 176 ppm, respectively. The other IIL peaks were marked and correlated to its chemical structure as represented in Figure 1b.

The chemical structures of IIL and Fe_3_O_4_-IIL were confirmed from their FTIR spectra summarized in Figure 2a,b, respectively. The disappearance of peaks at 2950 and 2850 cm^−1^ (Figure 2a; attributed to aliphatic C–H stretching) from the spectrum of Fe_3_O_4_-IIL and appearance of new peaks at 800 and 580 cm^−1^ (Figure 2b; attributed to Si–OH and Fe–O bending and stretching vibration, respectively) illustrate the formation of the Fe_3_O_4_-IIL chemical structure (Figure 1) [30]. The presence of an Si–O–Si bond was proved from the appearance of broad bands at 1024–1124 cm^−1^, 850 cm^−1^, which were attributed to Si–O–Si asymmetric and symmetric stretching vibration, confirming the hydrolysis of ethoxy groups of IIL and TEOS under basic condition. The appearance of bands at 3400, 1680, and 1580 cm^−1^ in the spectrum of IIL (Figure 2a; referred to NH stretching, C=O acetate stretching, and N–H bending vibration, respectively) confirms the formation of an imidazolium ring cation and acetate ion. The broad band at 3400 cm^−1^, referred to OH stretching, in the spectrum of Fe_3_O_4_-IIL proves the presence of an Si–OH and –OH group surrounding siloxane and iron oxide. Moreover, the lower intensity of the O–H band in the spectrum of Fe_3_O_4_-IIL (Figure 2b) confirms the linking of the hydroxyl groups surrounded on the magnetite surfaces with the hydroxyl groups of Si–OH during the hydrolysis of IIL in the presence of magnetite NPs [30]. These results elucidate that the magnetite NPs were linked with silica via the hydrolyzing of TEOS and triethoxy groups of IIL. The linking of magnetite with IIL increases its oxidation stability without the formation of other iron oxides such as hematite, maghemite, and others.

The thermal stability of IIL and Fe_3_O_4_-IIL and contents of magnetite and silica were evaluated from their thermogravimetric and differential thermogravimetric (TGA-DTG thermograms represented in Figure 3a,b, respectively. The initial degradation temperature (IDT) of IIL was started at 350 °C. The presence of magnetite and silica in the chemical structure Fe_3_O_4_-IIL loses approximately 8 wt. % at temperature ranged from 70 to 120 °C. These data confirm that the IIL cannot adsorb humidity; rather, the Fe_3_O_4_-IIL NPs have the tendency to adsorb the humidity on the magnetite and silica surfaces [32,33]. Figure 3 shows that the IIL starts to lose weight (approximately 15 wt. %) from about 220 °C and in the range 220 to 420 °C, which is attributed to the presence of acetate anion that degraded to DIW and CO_2_. The IDT of Fe_3_O_4_-IIL at 415 °C (Figure 3) confirms that the chemical linking of magnetite and siloxane improved its thermal stability and reduces the plasticization effect of IIL [28]. The remained residual after the decomposition of IIL and Fe_3_O_4_-IIL are 76 and 14 wt. % at 750 °C confirm that the magnetite and silica contents of Fe_3_O_4_-IIL are 48 and 12 wt. % (after excluding the humidity and remained residual of IIL).

The phase transitions of IIL and the interaction that occurred between and Fe_3_O_4_-IIL and pure IIL were determined from the DSC thermograms represented in Figure 4. The blending of Fe_3_O_4_-IIL with IIL (25 wt. %) occurred to show their interactions as reported for some magnetic IIL [28]. Their T_g_ and melting temperature (T_m_) were recorded on their curves (Figure 4). It was noticed that the presence of Fe_3_O_4_-IIL shifted the T_g_ and T_m_ values of IIL to suggest the existence of interactions among Fe_3_O_4_-IIL and pure IIL.

The morphology and particle sizes (nm) of Fe_3_O_4_-IIL were determined from TEM and DLS measurements, as illustrated in the experimental section and represented in Figure 5a,b, respectively. The morphology of Fe_3_O_4_-IIL (Figure 5a) show stretched rough spherical nanoparticles (NPs) with particle sizes at dry state that approximately ranged from 15 to 26 nm with the formation of small aggregates. The dispersion and particle sizes were increased when Fe_3_O_4_-IIL dispersed in DIW at pH 7 in 0.001 M of KCl (Figure 5b). Its polydispersity index and particle sizes from DLS data (Figure 5b) are 0.183 ± 0.01 and 31.2 ± 1.5 nm. The presence of an amino group in the chemical structure of IIL increases the basicity of the reaction medium and facilitates the rate of hydrolysis of TEOS with IIL. Moreover, the polarity of the produced Fe_3_O_4_-IIL changes with the formation of magnetite to stretch the spherical morphology of IIL due to the interaction of silica with magnetite [34]. The zeta potentials or surface charges (mV) of IIL and Fe_3_O_4_-IIL at pH 7 were measured and found to be −15.3 and −30.4 mV, respectively. The increasing of surface charges of IIL with the formation of silica and magnetite was referred to electron pair of amino group and acetate anion of IIL and the presence of hydroxyl groups of silica and magnetite [35]. These data confirm that the magnetite, silica, and amino groups of IIL (negative charges) oriented at the Fe_3_O_4_-IIL and imidazolium group were oriented into the interior core of the NPs. This means that the linking of hydroxyl groups of hydrolyzed siloxane group with hydroxyl groups of magnetite can be easily obtained in basic medium to form negatively dispersed Fe_3_O_4_-IIL.

The surface morphology of Fe_3_O_4_-IIL was investigated from the SEM micrograph represented in Figure 6. The SEM micrograph shows the spherical rough surface that agrees with the TEM and DLS data (Figure 5a,b). The rough surfaces can be attributed to the formation of silica and magnetite on the IIL surfaces on the basis of the added negative charge of the nanomaterials surface, which would enhance the dispersion of Fe_3_O_4_-IIL due to repulsive forces among the NPs [36].

The XRD diffractograms of IIL and Fe_3_O_4_-IIL were used to determine their lattice structures as summarized in Figure 7a,b. The IIL (Figure 7a) shows an amorphous lattice structure and Fe_3_O_4_-IIL (Figure 7b) suggests diffraction peaks that correspond to the (2 2 0), (3 1 1), (4 0 0), (4 2 2), (5 1 1), and (4 4 0) of magnetite and no obvious sharp diffraction peak corresponding to the IIL to indicate that most of the IIL is amorphous. These data confirm that the magnetite can be formed without other iron oxides during the hydrolysis of IIL in the basic condition to prove the strong capability of IIL as capping to protect the oxidation of magnetite to another iron oxide [30,31].

The magnetic properties of Fe_3_O_4_-IIL can be investigated from its VSM hysteresis loop as represented in Figure 8. The saturation magnetization (emu. g^−1^), remenant magnetization (emu. g^−1^), and coercivity (G) of Fe_3_O_4_-IIL are 33.41, 0.19, and 9.04, respectively. These values confirm the superparamagnetic characteristics of the prepared Fe_3_O_4_-IIL due to the lowering of coercivity and remnant magnetization and increasing of saturation magnetization when compared with other coated magnetite NPs [30,31,36]. These data confirm also that the magnetite NPS were capped on the Fe_3_O_4_-IIL surfaces as confirmed from TEM and DLS data (Figure 5a,b).

### 3.2. Curing of Fe_3_O_4_-IIL with Epoxy and Polyamine Hardener

The epoxy resin was cured with long-chain aliphatic polyamine (DGEB/PA) in the absence and presence of different weight ratios of Fe_3_O_4_-IIL (ranged from 1 to 6 wt. % related to DGEB/PA weights). The chemical structure of cured epoxy in the absence and presence of Fe_3_O_4_-IIL (6 wt. %) was selected and represented in Figure 9a,b, respectively. The intensity of bands at 3465, 1650, and 1121 cm^−1^, attributed to OH, C–N and C–O, or Si–O–Si stretching vibrations, was increased in the DGEB/PA-Fe_3_O_4_-IIL spectrum (Figure 9b) more than the cured DGEB/PA (Figure 9a). This result elucidates that the Fe_3_O_4_-IIL was incorporated to produce more hydroxyl groups in the polymer networks. The presence of new broad intense bands at 1650, 905, and 580 cm^−1^, referred to NH bending, Si–OH, and Fe–O stretching vibrations, confirms that imidazolium, siloxane, and magnetite NPs were incorporated with the epoxy without thermal degradation during the curing process.

The curing of the epoxy with long-chain aliphatic polyamine hardener is a very important parameter that controls the mechanical and anti-corrosion characteristics of the epoxy coatings. This can be attributed to the fast chemical curing producing microcracks and holes besides the increasing of epoxy toughness. It was previously reported that the incorporation of iron oxide and ILs during the curing of epoxy with hardener improves the thermal and mechanical properties of the epoxy resins [37]. In this respect, this work aims to modify the curing, thermal, and mechanical properties of the epoxy resin based on diglycidyl ether of bisphenol A cured with polyamine (DGEB/PA) in the presence different weight ratios of Fe_3_O_4_-IIL (ranged from 1 to 6 wt. % related to DGEB/PA weights). DSC investigated the curing behavior of DGEB/PA under non-isothermal mode at a heating rate of 5 °C min^−1^, as represented in their thermograms represented in Figure 10. This heating rate was selected as a slow heating rate to study the curing exothermic reactions of epoxy [37]. The total heat of the curing reactions of DGEB/PA as blank and mixed with different wt. % of Fe_3_O_4_-IIL were calculated from the area under peaks (Δ H) and summarized in Table 1. Careful inspection of the DGEB/PA curing data (Table 1 and Figure 10) proves that the addition of Fe_3_O_4_-IIL reduces the onset curing temperature (Table 1) and increases the curing reaction enthalpy. These data mean that the incorporation of magnetite and IIL (containing amino and imidazolium N atoms having a lone pair of electrons) accelerates the curing process of DGEB/PA. Accordingly, Fe_3_O_4_-IIL can act as a curing catalyst or co-curing agent with PA hardener. The presence of a shoulder peak in the curing curves of DGEB/PA at lower temperature in the presence 3 and 6 wt. % of Fe_3_O_4_-IIL (Figure 10) confirms the catalytic behavior of iron oxide as a curing catalyst [37,38]. The glass transition temperatures before curing and after dynamic curing represented as T_g1_ and T_g2_, respectively were measured for the DGEB/PA system in the absence and presence of Fe_3_O_4_-IIL and summarized in Table 1. The T_g1_ values determined for DGEB/PA blends on the cooling run before heating to show the effect of Fe_3_O_4_-IIL on the glass transition temperatures of DGEB/PA. Moreover, increasing the T_g1_ value of DGEB/PA (Table 1) with the incorporation of Fe_3_O_4_-IIL elucidates its good dispersion into an epoxy system before curing even with increasing its wt. % up to 6 wt. % [28]. The T_g2_ values were recorded from the second heating run at a heating rate of 5 °C min^−1^ after curing under nitrogen, as represented in Figure 10. The decreasing of T_g2_ with the incorporation of Fe_3_O_4_-IIL (Figure 10 and Table 1) confirms the good dispersion of Fe_3_O_4_-IIL into a DGEB/PA network and the flexibility of the network after curing due to the plasticizing effect of IIL. It was noticed that the T_g2_ decreased with increasing Fe_3_O_4_-IIL up to 3 wt. % and increased with increasing Fe_3_O_4_-IIL at 6 wt. % to confirm the lower dispersion of Fe_3_O_4_-IIL more than 3 wt. % in the DGEB/PA.

The surface morphologies and roughness of the cured epoxy DGEB/PA in the absence and presence different wt. % of Fe_3_O_4_-IIL NPs were investigated by SEM and AFM micrographs represented in Figure 11a–d and Figure 12a–d, respectively. It was noticed that the cured blank DGEB/PA shows micro cracks (Figure 11a and Figure 12a) due to the fast curing of the DGEB/PA system in the absence of Fe_3_O_4_-IIL NPs. The cracks among the epoxy layers and holes appeared in SEM of cured epoxy blank and in the presence of Fe_3_O_4_-IIL NPs 1 and 2 wt. % (Figure 11a–c). The incorporation of Fe_3_O_4_-IIL NPs (from 1 to 2 wt. %) started to fill the cracks and holes as appeared in Figure 11b,c and Figure 12b,c. The incorporation of Fe_3_O_4_-IIL NPs (3 wt. %) during the epoxy curing with PA produces uniform rough surfaces without holes or cracks (Figure 11 and Figure 12d). Figure 11d shows that the Fe_3_O_4_-IIL NPs (3 wt. %) fills all cracks to produce uniform layers. The high loading of Fe_3_O_4_-IIL NPs (6 wt. %) into DGEB/PA leads to formation of the aggregates or clusters (Figure 11 and Figure 12e).

### 3.3. DMA and Thermal Characteristics

The rigidity and flexibility of the cured epoxy networks based on DGEB/PA in the presence and absence of Fe_3_O_4_-IIL can be investigated from DMA as represented in Figure 13a,b. The glass transition temperatures of the cured epoxy resins in the presence and absence of Fe_3_O_4_-IIL determined from the effect of temperature on Tan δ (Figure 13a) agree with those obtained from DSC thermograms (Figure 10). Moreover, the onset temperatures for the storage modulus (Figure 13b) of DGEB/PA in the presence different wt. % of Fe_3_O_4_-IIL 0, 1, 2, 3, and 6 wt. % are 79.4, 66.8, 69.5, and 79.8 °C, respectively. These data prove that the mixing of DGEB/PA blank with Fe_3_O_4_-IIL (1–3 wt. %) increases the epoxy network flexibility and mobility. The stiffness of epoxy networks was increased by increasing Fe_3_O_4_-IIL NPs contents by more than 3 wt. %. This can be attributed to the effect of magnetite and ILs on the curing rate of epoxy crosslinking that increased with increasing the Fe_3_O_4_-IIL contents due to the formation of highly crosslinked heterogeneous epoxy networks [37]. Moreover, the increasing of Fe_3_O_4_-IIL NPs more than 3 wt. % decreases their dispersion in the epoxy matrix to produce clusters that increase the rigidity of the epoxy networks [38].

The crosslinking densities (ρ, mol. dm^−3^) are hard to ascertain for strong resistances of epoxy to swell in organic solvents. The crosslinking densities (ρ, mol. dm^−3^) of the cured DGEB/PA system in the absence and presence different wt. % of Fe_3_O_4_-IIL NPs networks can be determined as ρ = G_e_/*R*T_e_; where *R* = universal gas constant (8.314 J mol^−1^ K^−1^), *G*e is the storage modulus (MPa; after the samples have reached a plateau), and T_e_ is the extended temperature. The T_e_ values were determined from relation T_e_ = T_g_ + 30 °C (taken from the loss modulus data). The semi-empirical relation of ρ and G_e_ was more correctly applied to lightly crosslinked elastomers but it was previously applied in the curing of epoxy with ILs [21]. The data summarized in Table 2 confirm that the crosslinking densities of DGEB/PA (blank was increased in the order Fe_3_O_4_-IIL NPs wt. % 6 >1>3, which agrees with the T_g_ data determined with DSC (Figure 10) or DMA data (Figure 13a)). These data elucidate that the mixing of Fe_3_O_4_-IIL NPs with DGEB/PA produces an elastic network with increasing the Fe_3_O_4_-IIL NPs contents from 1 to 3 wt. %. This observation was referred to the presence of a higher number of initiating species being formed at the expense of ether linkages with the increase in Fe_3_O_4_-IIL NPs concentrations [21].

The effect of Fe_3_O_4_-IIL NPs on the thermal properties of the cured epoxy DGEB/PA in the absence and presence different wt. % of Fe_3_O_4_-IIL NPs was evaluated from TGA thermograms represented in Figure 14. The data of thermal stability proves that the increasing of Fe_3_O_4_-IIL NPs contents from 1 to 6 wt. % improves the thermal stability of the cured epoxy rather than that of the blank. It was also found that the blank DGEB/PA was degraded at temperature of 285 °C and 15 wt. % of the epoxy degraded up to 350 °C. The initial thermal degradation temperatures of DGEB/PA cured in the presence of 1, 3, and 6 wt. % of Fe_3_O_4_-IIL NPs were 290, 350, and 400 °C, respectively. It was also found that the char yields or residues at 650 °C were increased with increasing wt. % of Fe_3_O_4_-IIL NPs to confirm that the presence of magnetite and IIL leads to the formation of a barrier that prevents the evolution of volatiles during the degradation of organic compounds and increases the residues [39].

The previous DMA, TGA, and DSC data proposed that the Fe_3_O_4_-IIL NPs will initiate the polymerization of the epoxide ring at room temperature, which was confirmed via the appearance of a shoulder peak in the curing curves of DGEB/PA at lower temperature in the presence 3 and 6 wt. % of Fe_3_O_4_-IIL (Figure 10). It was previously reported that the curing of epoxy was carried out with 1-ethyl-3-methylimidazolium acetate, 1-ethyl-3-methylimidazolium diethyl phosphate, 1-ethyl-3-methylimidazolium dicyanamide, and 1-ethyl-3-methylimidazolium thiocyanate via either a nucleophilic route or the decomposition of IILs via the carbine route and imidazole route with anionic polymerization [21]. The presence of an amine group in the chemical structure of IIL improves the nucleophilic ring opening of epoxy, as illustrated in Figure 2. Moreover, the presence of acetate anion facilitates the carbene reaction of imidazolium cation that was produced by the deprotonation of imidazolium to carbene with the formation of acetic acid [37]. The imidazolium carbene is responsible to start an acid-aided epoxy–amine curing reaction with the aid of magnetite NPs [37].

## 4. Conclusions

New IIL based on 2-(4-minophenyl)-1.3-bis(triethoxysilyl)-1H-imidazol-3-ium acetate was hydrolyzed with TES and iron cations to produce Fe_3_O_4_-IIL NPs. The uniform Fe_3_O_4_-IIL NPs nanostructures were confirmed from its sizes, polydispersity index, and surface charges as 31.2 nm, 0.183 ± 0.01 nm, and 31.2 ± 1.5 nm, −30.4 mV, respectively. The DMA, DSC, TGA, SEM, and AFM data confirm that the incorporation of Fe_3_O_4_-IIL during the curing of DGEB/PA increases the epoxy networks flexibility and thermal stability after curing due to the plasticizing effect of IIL. The glass transition temperatures of the cured DGEB/PA determined from DSC data in the presence of different wt. % of Fe_3_O_4_-IIL 0, 1, 3, and 6 wt. % are 105, 95, 85, and 98, respectively. Moreover, the onset temperatures for the storage modulus of DGEB/PA in the presence different wt. % of Fe_3_O_4_-IIL 0, 1, 2, 3, and 6 wt. % are 79.4, 66.8, 69.5 and 79.8 °C, respectively. These data prove that the mixing of a DGEB/PA blank with Fe_3_O_4_-IIL (1–3 wt. %) increases the epoxy network’s flexibility and mobility. It was also concluded that Fe_3_O_4_-IIL (3 wt. %) was sufficient to modify the curing of the DGEB/PA epoxy matrix.

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
