# Peer review of "Magnetic Ionic Liquid Nanocatalyst to Improve Mechanical and Thermal Properties of Epoxy Nanocomposites"

_nanomaterials, 2020, doi:10.3390/nano10122325_

Round 1

Reviewer 1 Report

General comments:

In this work a quite complete and deserves its publication. The amount of experimental work is enough, however the comparison of the experimental data vs. those from the literature must be presented in a more proper manner while the discussion of the results and the conclusions made are adequate to recommend this paper. Finally, I do think that this paper can be very useful to researchers working in this research field, especially if authors improve the overall discussion of their main manuscript by highlighting main advantages vs. drawbacks of their strategy vs. those already reported in the literature.  

Recommendation:

To sum up, this paper can be published after some a major revision taking into account all comments listed below.

Major changes requested:

C1. Introduction: Authors must report briefly a state of the art of data for other MIL (Magnetic Ionic Liquids) available into the literature. As is, the introduction is too short. Could authors explain in greater detail main difference between their strategy vs. those already reported in the literature?

C2. Experimental Section: The calibration of each equipment must be explained in detail. Then, the accuracy and the uncertainty in each reported property need to be more clearly quantified (including also uncertainty in the temperature, pressure and composition). And table presenting experimental data must be completed by adding uncertainty in each reported quantity.

C3. Purity of their IIL sample must be provided by including water, halide and metal content along with all synthesis steps. Furthermore, CHNS and halide content must be determined to fully characterize each IL sample. Could you also provide in the ESI at least NMR (13C and 2D NMR) in addition to 1H NMR spectrum provided?

C4. Authors must investigate the thermal stability of selected materials in greater detail: DSC traces must be presented in an ESI. Furthermore, static TGA analysis must be run to reflect truly the stability of their IL/IIL as a function of time for a given temperature.

C5. Quality of each graph must be improved.

Author Response

Major changes requested:

C1. Introduction: Authors must report briefly a state of the art of data for other MIL (Magnetic Ionic Liquids) available into the literature. As is, the introduction is too short. Could authors explain in greater detail main difference between their strategy vs. those already reported in the literature?

Answer: New references and sentences marked with the red colour were inserted to improve the introduction section.

C2. Experimental Section: The calibration of each equipment must be explained in detail. Then, the accuracy and the uncertainty in each reported property need to be more clearly quantified (including also uncertainty in the temperature, pressure and composition). And table presenting experimental data must be completed by adding uncertainty in each reported quantity.

Answer: The experimental section was modified with new paragraph and new sentences marked with the red color. The conditions used for samples analyses were also inserted in the text. All analyses conditions were reported.

C3. Purity of their IIL sample must be provided by including water, halide and metal content along with all synthesis steps. Furthermore, CHNS and halide content must be determined to fully characterize each IL sample. Could you also provide in the ESI at least NMR (13C and 2D NMR) in addition to 1H NMR spectrum provided?.

Answer: The IIL ddoes not contain any halides it is based on acetate. The nitrogen content added in the preparation section to confirm its purity. we have not ability to do 2D NMR for more clarification 13CNMR spectrum inserted Figure 1b.

C4. Authors must investigate the thermal stability of selected materials in greater detail: DSC traces must be presented in an ESI. Furthermore, static TGA analysis must be run to reflect truly the stability of their IL/IIL as a function of time for a given temperature.

Answer: The phase transitions of IIL and the interaction occurred between and Fe3O4-IIL and pure IIL were determined from DSC thermograms represented in Figure . The blending of Fe3O4-IIL with IIL (25 Wt.%) was occurred to show their interactions as reported for some magnetic IIL [ ref]. Their Tg and melting temperature (Tm) were recorded on their curves (Figure ). It was noticed that the presence of Fe3O4-IIL shifted the Tg and Tm values of IIL to suggest the existence of interactions among Fe3O4-IIL and pure IIL.

Reviewer 2 Report

In this work, Fe3O4-IIL nannoparticles were mixed with epoxy/polyamine systems to improve the curing, thermal and mechanical properties of epoxy, is studied

Paragraph 3.1 is dedicated to the evaluation of thermal stability, morpholgy, etc., to confirm the formation of uniform, thermal stable and superparamagnetic Fe3O4-IIL Nps. I understand that these results, that are sumarized in Figures 1 to Figure 7, are not the scope of the paper but an intermediate step to achive the improvement of the properties of the epoxy by means of the addition of the Nps. I have no objections about his part. Perhaps is too long and, of course, the figures must be improved. They are not homogeneous (compare, for example, the y-label in figure 6 with the one in figure 3)

For me, paragrapah 3.2 is the core of the paper. In this sense, Figure 9 (there is a mistake in line 261, pp 9, it is not figure 8), Figure 12 and Figure 13 are very important. The same can be said for Table 1 and Table 2. I think that the numerical results in the tables must be clearly related to the figures. For example, I do not see how the glass transition before and after curing are obtained, or the area under the peaks. The same for the storage modulus (the figure is in GPa but the value is in MPa). I do not find the meaning of the temperatures in Figure 12.b. Why to exprress Te in K? Where is the loss modulus (that the authors used to calculate Te). But these are only some examples.

In summary, I think that the authors have to improve the explanation of the numerical values obtained from the experimental results showed in Figures 9, 12 and 13, before this paper can be published.

Author Response

Comments and Suggestions for Authors

Paragraph 3.1 is dedicated to the evaluation of thermal stability, morpholgy, etc., to confirm the formation of uniform, thermal stable and superparamagnetic Fe3O4-IIL Nps. I understand that these results, that are sumarized in Figures 1 to Figure 7, are not the scope of the paper but an intermediate step to achive the improvement of the properties of the epoxy by means of the addition of the Nps. I have no objections about his part. Perhaps is too long and, of course, the figures must be improved. They are not homogeneous (compare, for example, the y-label in figure 6 with the one in figure 3)

Answer:It is necessary to interpretate  the analyses data of new prepared IIL to confirm its characteristics as capping for magnetite nanomaterials. It is not intermediate it is final product that should be characterized as recommended to be more discussion with other reviewer.

For me, paragrapah 3.2 is the core of the paper. In this sense, Figure 9 (there is a mistake in line 261, pp 9, it is not figure 8), Figure 12 and Figure 13 are very important.

Answer: The figure number was corrected.

The same can be said for Table 1 and Table 2. I think that the numerical results in the tables must be clearly related to the figures. For example, I do not see how the glass transition before and after curing are obtained, or the area under the peaks.

Answer: The intergration of peak using linear base line yields the heat of curing that summarized in Table 1. Onset temperatures cannot recorded on figures it was recorded for every thermogram and represented in table 1. The Tg1 values determined for DGEB/PA blends on cooling run before heating to show the effect of Fe3O4-IIL on the glass transition temperatures of DGEB/PA. Moreover, increasing Tg1 value of DGEB/PA (Table 1) with incorporation of Fe3O4-IIL elucidates its good dispersion into epoxy system before curing even with increasing its Wt.% up to 6 Wt.% [34]. The Tg2 values were recorded from the second heating run at heating rate 5 oC.min-1 after curing under nitrogen as represented in Figure 10. The decreasing of Tg2 with the incorporation of Fe3O4-IIL (Figure 10 and Table 1) confirm the good dispersion of Fe3O4-IIL into DGEB/PA network and flexibility of the network after curing due to plasticizing effect of IIL. It was noticed that The Tg2 decreased with increasing Fe3O4-IIL up to 3 Wt. % and increased with increasing Fe3O4-IIL at 6 Wt. % to confirm the lower dispersion of Fe3O4-IIL more than 3 Wt.% in the DGEB/PA.  

The same for the storage modulus (the figure is in GPa but the value is in MPa). I do not find the meaning of the temperatures in Figure 12.b. Why to exprress Te in K? Where is the loss modulus (that the authors used to calculate Te). But these are only some examples.

Answer: The values recorded on the logarithmic plot of storage modulus recorded in GPa while the values used for calculation of the crosslinking density: ρ = Ge/RTe required both Te and G units should be K and MPa.

Reviewer 3 Report

The MS entitled „Magnetic Ionic Liquid NanoCatalyst to Improve

Mechanical and Thermal Properties of Epoxy Nanocomposites“ written by Ayman M. Atta, Ahmed M. Tawfeek, and Nourah I. Sabeela focusses on the synthesis of the ionic liquid 2-(4-aminophenyl)-1.3-bis(triethoxysilyl)-1H-imidazol-3-ium acetate for manufacturing of Fe3O4 nanoparticles and application of the latter in an epoxy polyamine curing system. Structural analysis of the cured materials obtained was carried out using FT IR spectroscopy, TEM, DLS, SEM, XRD, DSC, AFM, DMA, and TGA analysis. The MS is interesting, however some questions exist.

Conditions are missing for sonication. Authors gave only temperature and time, however, conditions, such as reaction volume and amplitude used for sonication are missing in the MS.

Furthermore, aqueous medium is used for ionic liquid preparation. To which extent hydrolysis of the silane group does occur under these conditions? Basic conditions were applied later as well. Do the basic conditions affect the imidazolium structure?

What do authors mean with amine salt? This name may be something confusing.

The weight loss of approximately 8 wt % in the TGA curves discussed between 70°C and 120°C may be caused by remaining volatiles. How do these volatiles influence the glass transition temperature of the cured material?

On page9 of the MS line 16 from bottom: it must read Figure 9 for the DSC instead of Figure 8. Furthermore, it is written “…amino and imidazolium N atom with a lone pair”. However, imidazolium is an aromatic system. What do the authors call lone pair when they discuss imidazolium ion?

Usually glass transition temperatures are given as full numbers.

Figure 12 contains comparison of DMA data. Why is the curve of the sample containing 6 wt % of Fe3O4 nanoparticles more close to the blank although samples with 1 wt % and 3 wt % significantly differ from the curve of the blank? A detailed explanation is missing for this result. How do the Fe3O4 nanoparticles influence the crosslinking density of the material?

The conclusions need improvement as well because it reads as a summary.

I cannot recommend the MS for publication in the present form. Major revision is necessary.

Author Response

Conditions are missing for sonication. Authors gave only temperature and time, however, conditions, such as reaction volume and amplitude used for sonication are missing in the MS.

Answer: The experimental section was modified with new paragraph and new sentences marked with the red color. The conditions used for samples analyses were also inserted in the text. All analyses conditions were reported.

Furthermore, aqueous medium is used for ionic liquid preparation. To which extent hydrolysis of the silane group does occur under these conditions? Basic conditions were applied later as well. Do the basic conditions affect the imidazolium structure?

Answer: Deionized water (DIW) with 0.1 MΩ cm resistivity was used for preparing IIL. The basic conditions are necessary to prepare both magnetite and siloxane hydrolysis in ethanol/water and basic conditions.

What do authors mean with amine salt? This name may be something confusing.

Answer: The formation of amine saltof ABA( CH3COOH.NH2-Ph) at temperature (-4 oC) prevents the condensation of amine group with glyoxal and facilitates the condensation of its aldehyde group with APTS.

The weight loss of approximately 8 wt % in the TGA curves discussed between 70°C and 120°C may be caused by remaining volatiles. How do these volatiles influence the glass transition temperature of the cured material?

Answer: The conditions for analyses were inserted in the experimental section confirm that all materials were dried before thermal analyses to delay the effect of humidity on the accuracy of analyses.

On page9 of the MS line 16 from bottom: it must read Figure 9 for the DSC instead of Figure 8. Furthermore, it is written “…amino and imidazolium N atom with a lone pair”. However, imidazolium is an aromatic system. What do the authors call lone pair when they discuss imidazolium ion?

Answer: The presence of amine group in the chemical structure of IIL improves the nucleophilic ring opening of epoxy as illustrated in Scheme 2. IIL has primary amine group.

Figure 12 contains comparison of DMA data. Why is the curve of the sample containing 6 wt % of Fe3O4 nanoparticles more close to the blank although samples with 1 wt % and 3 wt % significantly differ from the curve of the blank? A detailed explanation is missing for this result. How do the Fe3O4 nanoparticles influence the crosslinking density of the material?

Answer: These data elucidate that the mixing of Fe3O4-IIL NPs with DGEB/PA produces elastic network with increasing the Fe3O4-IIL NPs contents from 1 to 3 Wt. %. This observation was referred to the presence of a higher number of initiating species being formed at the expense of ether linkages with the increase in Fe3O4-IIL NPs concentrations [40]. The presence of acetate anion facilitates the carbene reaction of imidazolium cation that produced by deprotonation of imidazolium to carbene with formation of acetic acid . The imidazolium carbene is responsible to start acid-aided epoxy-amine curing reaction with aid of magnetite NPs.

Answer: The conclusion rewrote.

Round 2

Reviewer 1 Report

Many Thanks for sharing this improved version of your mas which is now ready to be published.

Thanks

Reviewer 2 Report

I believe that the manuscript has been significantly improved and now warrants publication in Nanomaterials